# Microstructure and Ablation Behavior of C/C-SiC-(Zr_x_Hf_1−x_)C Composites Prepared by Reactive Melt Infiltration Method

**DOI:** 10.3390/ma16052120

**Published:** 2023-03-06

**Authors:** Zaidong Liu, Yalei Wang, Xiang Xiong, Zhiyong Ye, Quanyuan Long, Jinming Wang, Tongqi Li, Congcong Liu

**Affiliations:** 1State Key Laboratory of Powder Metallurgy, Central South University, Changsha 410083, China; 2Science and Technology of Advanced Functional Composites Laboratory, Aerospace Research Institute of Materials and Processing Technology, Beijing 100076, China

**Keywords:** ceramic matrix composites, microstructure, ablation behavior, reactive melt infiltration

## Abstract

C/C-SiC-(Zr_x_Hf_1−x_)C composites were prepared by the reactive melt infiltration method. The microstructure of the porous C/C skeleton and the C/C-SiC-(Zr_x_Hf_1−x_)C composites, as well as the structural evolution and ablation behavior of the C/C-SiC-(Zr_x_Hf_1−x_)C composites, were systematically investigated. The results show that the C/C-SiC-(Zr_x_Hf_1−x_)C composites were mainly composed of carbon fiber, carbon matrix, SiC ceramic, (Zr_x_Hf_1−x_)C and (Zr_x_Hf_1−x_)Si_2_ solid solutions. The refinement of the pore structure is beneficial to promote the formation of (Zr_x_Hf_1−x_)C ceramic. The C/C-SiC-(Zr_x_Hf_1−x_)C composites exhibited outstanding ablation resistance under an air–plasma environment at around 2000 °C. After ablation for 60 s, CMC-1 appeared to possess the minimum mass and linear ablation rates of only 2.696 mg/s and −0.814 µm/s, respectively, which are lower than those of CMC-2 and CMC-3. During the ablation process, a Bi-liquid phase and a liquid–solid two-phase structure were formed on the ablation surface which could act as an oxygen diffusion barrier to retard further ablation, which is responsible for the excellent ablation resistance of the C/C-SiC-(Zr_x_Hf_1−x_)C composites.

## 1. Introduction

There is fierce competition in the space exploration field, bringing about increasing demands for advanced and innovative structural materials for future space systems [1,2]. Carbon/carbon (C/C) composites possess outstanding characteristics of low density, high thermal conductivity, low coefficient of thermal expansion, high specific strength and so on, which translates into appreciable application value in the aerospace industry [3,4,5]. However, the rapid oxidation of C/C composites strongly limits their practical applications in extreme environments with high-pressure airflow and hyperthermal flux [6,7]. Therefore, it is urgent to improve the ablation resistance of C/C composites in harsh high-temperature oxidation atmospheres.

Many studies have confirmed that it is a favored route to improve the ablation resistance of C/C composites by adding ultra-high temperature ceramics (UHTCs), such as zirconium carbide (ZrC) [8,9], hafnium carbide (HfC) [10,11,12] or tantalum carbide (TaC) [13,14]. Among these UHTCs, HfC and ZrC possess extremely high melting points (3890 °C and 3540 °C, respectively), good ablation resistance and thermal stability [9,15]. Xiong et al. [16] prepared a HfC/ZrC ablation protective coating for C/C composites by chemical vapor deposition, and the results showed that the HfC/ZrC ceramic coating exhibited outstanding ablation resistance under oxyacetylene flame. It is well known that hafnium oxide (HfO_2_) and zirconia (ZrO_2_) as oxidation products of HfC and ZrC possess high melting points and relatively low evaporation pressures [17], which are responsible for the outstanding ablation resistance of HfC and ZrC ceramics. Unfortunately, both HfO_2_ and ZrO_2_ are usually formed with porous structures during the lower temperature oxidation process. The porous structures can provide diffusion channels for oxygen to enter the composites, which is contrary to the ablation performance of C/C-UHTCs composites [18]. In order to promote the ultra-high-temperature ablation performance of C/C-UHTC composites, the introduction of a multiphase UHTCs matrix in C/C composites could be a feasible approach [19,20]. SiC possesses outstanding oxidation resistance at moderate temperature. Glassy SiO_2_ products can effectively retard the further infiltration of oxygen and reduce oxidation-caused mass loss during ablation [21,22]. In addition, HfO_2_ and ZrO_2_ can dissolve into liquid SiO_2_ to form a Si-Zr-Hf-O glassy phase under higher temperatures, which can effectively work as a binder to hold the HfO_2_/ZrO_2_ particles onto the protective layer [12,23]. The interactions among the above oxides can increase the viscosity of the glassy phases, and then reduce the risk of solid oxides being rinsed away by high-speed airflow [24]. Therefore, it is an effective method to achieve the improvement of the ablation resistance of C/C composites by introducing multicomponent carbide ceramics. Up to the present, the C/C-UHTC composites are usually prepared by chemical vapor infiltration (CVI) [25,26], precursor infiltration and pyrolysis (PIP) [27,28], slurry infiltration (SI) [29,30] and reactive melt infiltration (RMI) [23,31] methods, in which RMI is an effective method to prepare C/C-UHTC composites with the advantages of low cost, short period, near net shape and so on. Chen et al. [18] prepared C/HfC-ZrC-SiC composites with various HfC contents by the RMI method and investigated the effect of HfC content on the ablation resistance of the composites. The results show that the increased HfC content significantly improved the ablation resistance. However, there are few systematic studies on the ablation resistance of C/C-SiC-ZrC-HfC composites. In addition, the internal relevance between the microstructure and ablation behavior of C/C-SiC-ZrC-HfC composites remains unknown.

In this work, porous C/C composites with various densities were used to prepare C/C-SiC-(Zr_x_Hf_1−x_)C composites by the RMI method. The effect of the connected pore structure on the microstructure and ablation resistance of the C/C-SiC-(Zr_x_Hf_1−x_)C composites were investigated. The ablation behavior and mechanism were also discussed in detail.

## 2. Experimental

### 2.1. Material Preparation

For the experiments, 2.5D needled carbon fiber felts with a bulk density of 0.55 g/cm^3^ were used as preform for fabricating porous C/C composites by the CVI method. The infiltration temperature was controlled to be 950 ± 5 °C and the system pressure was in the range of 0.6–1.1 kPa. The propylene was used as the carbon precursor with a flow of 30 L/min, and the nitrogen was used as dilution gas with a flow of 10 L/min. In the present work, the porous C/C composites were designed with various densities, which was achieved by adjusting the infiltration duration. The final densities of porous C/C composites were examined to be 1.08 g/cm^3^, 1.25 g/cm^3^ and 1.35 g/cm^3^, which were denoted as C/C-1, C/C-2 and C/C-3, respectively.

Si (≤74 μm), Zr (≤44 μm) and Hf (≤44 μm) powders with the molar fractions of 70.00%, 17.15% and 12.85%, respectively, were used as the raw materials to prepare the infiltration material. The infiltration materials were prepared by ball milling for 24 h in alcohol medium and then drying in a vacuum oven for 24 h. Subsequently, the porous C/C composites were embedded in the as-prepared infiltration material in a SiC-coated graphite crucible. After the RMI process in a high-temperature furnace with argon atmosphere, C/C-SiC-(Zr_x_Hf_1−x_)C composites with different densities were successfully prepared. The infiltration temperature was set to be 2000 °C and the soaking time was 3 h. Corresponding to the porous C/C composites, the as-obtained C/C-SiC-(Zr_x_Hf_1−x_)C composites were denoted as CMC-1, CMC-2 and CMC-3, respectively.

### 2.2. Characterization

The bulk densities and porosities of the porous C/C and C/C-SiC-(Zr_x_Hf_1−x_)C composites were measured with the standard Archimedes method. The porous C/C composites with a size of 7 mm × 7 mm × 7 mm were measured by mercury intrusion porosimetry (MIP, Auto Pore IV 9500, Micromeritics, Norcross, USA) to quantify the pore size and distribution. The identified pore size was in the range of 0.006–300 μm and the pressure of the liquid mercury was gradually applied, from 0.2 to 30,000 psia. The porous C/C composites were scanned by 3D X-ray computer tomography (X-CT, Xradiation620 Versa, Zeiss, Oberkochen, Germany) with a 4× objective detector, and then the structural characteristics and parameters of the connected pores were obtained through the 3D reconstruction using Avizo 2019.1 software with a voxel size of 5 μm. The phase compositions of the C/C-SiC-(Zr_x_Hf_1−x_)C composites before and after ablation were investigated by X-ray diffraction (XRD, Advance-D8, Cu Kα1, Bruker, Billerica, Germany). The acceleration voltage and emission current were 40 kV and 40 mA, respectively, and the scan speed was 8°/min. The microstructure of the porous C/C and the C/C-SiC-(Zr_x_Hf_1−x_)C composites was investigated by scanning electron microscopy (SEM, Mira4, Tescan, Brno, Czech), and the element distributions were examined by energy dispersive spectroscopy (EDS, Xplore30. Aztec one, Oxford Instruments, UK).

### 2.3. Ablation Test

The ablation properties of the C/C-SiC-(Zr_x_Hf_1−x_)C composites were tested by an air-plasma equipment, as shown in Figure 1. The as-prepared CMC composites were machined into a cylinder shape (Φ30 mm × 9 mm) in an axis perpendicular to fabric, and then fixed in a sample holder coupled with a cooling system. The inner diameter of the gun tip was 4 mm. The pressure of argon and hydrogen were 5.0 bar and 3.3 bar, respectively. The flow rate of argon and hydrogen were 2000 L/h and 150 L/h, respectively. During ablation process, the ablation flame was parallel to the axial direction of the CMC samples, and the distance between the gun tip and the samples was set to be 60 mm. The surface temperature of the CMC samples was measured by an infrared thermometer. After ablation for 60 s, the mass and linear ablation rates were calculated using the following equations:(1)$$R_{m}=\frac{∆m}{t},$$
(2)$$R_{l}=\frac{∆l}{t},$$
where $R_{m}$ is the mass ablation rate, mg/s; ∆m is the mass loss, mg; $R_{l}$ is the linear ablation rate, μm/s; ∆l is the maximum thickness change at the ablation center, μm and t is the ablation time, s.

## 3. Results and Discussion

### 3.1. Microstructure of Porous C/C Composites

Figure 2 shows the cross-section SEM micrographs and pore size distributions of the porous C/C composites. It can be seen that the porous C/C composites present similar structural features, which can be roughly divided into non-woven-layer region, web-layer region and needled region. Evidently, the small-scale pores are mainly distributed inside the fiber bundles of the non-woven-layer and needled regions. The web-layer regions and the boundaries between neighboring regions act as enrichment areas for large-scale pores. In addition, a few large-scale pores can be found between the adjacent fiber bundles in each non-woven-layer region. As a result, the above pores built a complicated network structure in the porous C/C skeleton, which is critical for the introduction of Si-Zr-Hf melt during the infiltration process. Moreover, there are evident differences in the pore size of the porous C/C composites with various densities. Figure 2a shows the cross-section morphology of C/C-1. It can be seen that the deposition of pyrocarbon (PyC) led to high densification in the non-woven-layer and needled regions due to the high content and tight arrangement of the carbon fibers. On the contrary, a significant number of residual pores can be clearly observed in the other regions. With the density increased to be 1.25 and 1.35 g/cm^3^, the fiber bundles were almost full of PyC matrix, only some tiny pores can be observed, as shown in Figure 2b,c. Meanwhile, the pore size and porosity gradually decreased with the elevated density of C/C composites. Figure 2d shows the pore size distribution of the porous C/C composites. It is undoubted that the porous C/C composites present obvious unimodal distribution characteristics, although a faint peak can be detected in the range of 2–5 μm. The analysis shows that the pore size is mainly in the range of 5–100 μm, and the most probable pore size is about 31, 30 and 26 μm for C/C-1, C/C-2 and C/C-3, respectively. In addition, the reduced integral area represents the decrease volume of open pores in the porous C/C composites. The shift in peaks indicates that the introduction of the PyC matrix is mainly achieved by filling the larger-scale pores, which has been confirmed in Figure 2a–c. Based on the results from the Archimedes method, the open porosities of the porous C/C composites were 41.39 vol.%, 32.71 vol.% and 26.83 vol.%. The PyC content within C/C-1, C/C-2 and C/C-3 were calculated to be 25.48 vol.%, 33.65 vol.% and 38.46 vol.%, respectively, by the following equation:(3)$$V_{PyC}=\frac{\rho_{C/C}-\rho_{pre}}{\rho_{PyC}},$$
where $\rho_{C/C}$, $\rho_{pre}$ and $\rho_{PyC}$ are the densities of porous C/C composites, carbon preforms and PyC, in which $\rho_{pre}$ and $\rho_{PyC}$ are 0.55 g/cm^3^ and 2.08 g/cm^3^, respectively.

In order to better recognize and quantify the connectivity of the pores, the porous C/C composites were investigated by X-CT. Figure 3 shows the 3D reconstruction graphs of the connected pores for the porous C/C composites with various densities, in which the blue solid part is respective to the actual space occupied by the connected pores. Qualitative analysis shows that the connected porosities of the porous C/C composites are 40.66 vol.%, 31.72 vol.% and 26.66 vol.% for C/C-1, C/C-2 and C/C-3, respectively, which is overall consistent with the results from the Archimedes method. Although the connected pores seem to be complex and difficult to identify, the periodic distribution of the connected pores can be observed visually from the 3D graphs. Along with the z direction, pore-enrichment and pore-barren areas exist alternatively, which correspond to the web-layer and non-woven-layer regions, respectively. The representative pore structures were cut from the parent graph and displayed on the right, as shown in Figure 3. It can be seen that the connected pores in the non-woven-layer regions possess evident directivity, and slender, needle-like pores can be observed along the fiber orientation. These pores possess evident small-scale characteristics. Moreover, the large-scale pores in the web-layer regions exhibit outstanding connectivity. It can be found by comparison that the fragmented degree increases significantly with the elevated density of the porous C/C composites, which suggests that the introduction of the PyC matrix can play an effective role in the refinement of large-scale pores. The refinement effect can also be confirmed by the shift in peaks shown in Figure 2d.

Based on the 3D reconstruction results, Avizo software was used to calculate the fractal dimension and tortuosity of the connected pores in the porous C/C composites, in which the z direction was considered. Table 1 presents the structural parameters of the connected pores in the porous C/C composites. It can be seen that the fractal dimension is 2.68 for the C/C-1, while it is calculated to be 2.62 for both C/C-2 and C/C-3. Applied to 3D images, the fractal dimension is a rather effective indicator to measure and compare the roughness of a surface, which is in the range of 2-3. The larger the fractal dimension, the more irregular the pores in the porous C/C composites. The lower fractal dimension of the C/C-2 and C/C-3 composites may signify more effective infiltration for the Si-Zr-Hf melt in the porous C/C composites. In addition, relevant studies [32,33] show that the low tortuosity is beneficial to the infiltration of fluid. It can be found from Table 1 that the tortuosity is 1.73 for C/C-1, while those of C/C-2 and C/C-3 are 1.85 and 1.88, respectively. The corresponding changes mean the extension of infiltration paths and the refinement of connected pores in the porous C/C composites with the increased density.

### 3.2. Microstructure of C/C-SiC-(Zr_x_Hf_1−x_)C Composites

Figure 4 shows the XRD patterns of the CMC composites prepared by the RMI method. It can be seen that the characteristic diffraction peaks of the C, SiC, ZrC, HfC, ZrSi_2_ and HfSi_2_ phases can be identified according to the standard PDF cards, and there are no visible differences in the phase composition of the different CMC composites. The XRD analysis indicates that SiC, ZrC and HfC are the main phases in the CMC composites, which resulted from the chemical reaction between the PyC matrix and the Si-Zr-Hf melt. The small diffraction peaks of ZrSi_2_ and HfSi_2_ indicate the formation of alloys and the inadequate reactions between the Si-Zr-Hf melt and the PyC matrix during the RMI process. It is worth noting that the diffraction peaks of ZrC and HfC, as well as those of ZrSi_2_ and HfSi_2_, are difficult to distinguish from each other, which is attributed to the complete solid solution of Zr and Hf due to the similar lattice constant, atomic radius and electronegativity [34]. Therefore, it can be inferred that the (Zr_x_Hf_1−x_)C and (Zr_x_Hf_1−x_)Si_2_ solid solutions were formed during the RMI process [35,36,37], and that the C/C-SiC-(Zr_x_Hf_1−x_)C composites were composed of carbon fiber, carbon matrix, (Zr_x_Hf_1−x_)C and (Zr_x_Hf_1−x_)Si_2_ solid solutions. Moreover, with the elevated density of the porous C/C composites, the C content exhibits an increasing trend, as shown in Figure 4. On the contrary, the (Zr_x_Hf_1−x_)Si_2_ content decreases significantly. As mentioned above, the large-scale pores were gradually refined with the increased density of the C/C composites. Thus, it can be inferred that the shortened path for C atoms diffusion to the Si-Zr-Hf melt is responsible for the decreased (Zr_x_Hf_1−x_)Si_2_ content in C/C-2 and C/C-3.

Figure 5 shows the SEM micrographs of the C/C-SiC-(Zr_x_Hf_1−x_)C composites. It can be seen from the cross-section view that the CMC composites display visible structural heritability with that of the porous C/C composites. As shown in Figure 5a–c, the periodic non-woven-layer regions can be distinguished clearly. Meanwhile, the connected pores in the porous C/C composites have been filled with ceramics and alloy phases. The as-prepared CMC composites exhibit high density, which indicates the adequate infiltration of the Si-Zr-Hf melt into C/C composites during the RMI process. In the web-layer regions, except for the residual carbon, three kinds of phases can be detected according to the contrast difference. The classic structural characteristics are shown in the insets of Figure 5a. The EDS result in Figure 5g shows that the dark-grey phase is mainly composed of C and Si elements with corresponding mole fractions of about 52.45% and 47.50%, which confirm the formation of the SiC phase. In the same way, the light-grey phase and white phase are identified as (Zr_x_Hf_1−x_)Si_2_ solid solution and (Zr_x_Hf_1−x_)C solid solution, respectively, based on the XRD results (Figure 4) and EDS results (Figure 5h,i). Comprehensive analysis shows that all of the C/C-SiC-(Zr_x_Hf_1−x_)C composites are composed of the same components and possess similar structural characteristics. However, there are rather significant differences in the material structure of the C/C-SiC-(Zr_x_Hf_1−x_)C composites obtained from various porous C/C composites. It can be seen from Figure 5a that SiC, (Zr_x_Hf_1−x_)Si_2_, (Zr_x_Hf_1−x_)C and residual carbon are intermittently distributed in the web-layer regions of CMC-1. It is worth noting that the carbon fiber and PyC matrix have been almost exhausted, and the inadequate carbon sources are responsible for the high (Zr_x_Hf_1−x_)Si_2_ content. Meanwhile, a few of the (Zr_x_Hf_1−x_)C phases are homogeneously distributed around the (Zr_x_Hf_1−x_)Si_2_ phases, which is also a result of the lack of carbon source. Moreover, the metal Si in the raw infiltration material has been confirmed to be completely consumed by reacting with carbon fiber, the PyC matrix, metal Zr and Hf. For C/C-2 and C/C-3, the PyC matrix in the web-layer regions contributes the major increments in density, and there is a significant increase in carbon source, while the reduced pore size is beneficial to shorten the diffusion distance of carbon atoms to the Si-Zr-Hf melt. As a result, the transformational degree from (Zr_x_Hf_1−x_)Si_2_ to (Zr_x_Hf_1−x_)C significantly increases, resulting in the rapid consumption of (Zr_x_Hf_1−x_)Si_2_ and the formation of (Zr_x_Hf_1−x_)C, as shown in Figure 5b,c. Consequently, it can be confirmed that small-scale connected pores can effectively promote the ceramic transformation process. More residual carbon can be observed in the web-layer regions for CMC-2 and CMC-3, which may be unfavorable to the ablation resistance of C/C-SiC-(Zr_x_Hf_1−x_)C composites. Figure 5d–f shows the surface morphologies of C/C-SiC-(Zr_x_Hf_1−x_)C composites focused on the web-layer regions with enriched ceramics. It can be seen that the visible surfaces of different C/C-SiC-(Zr_x_Hf_1−x_)C composites are mainly occupied by ceramics and alloy phases. Only a small amount of PyC and carbon fibers are exposed. In addition, the component phases are unevenly distributed on the surface, and the changes in phase composites are roughly consistent with those observed from the cross-section of the web-layer regions.

After the RMI process, the densities and porosities of the prepared C/C-SiC-(Zr_x_Hf_1−x_)C composites exhibited a slight difference with each other. CMC-1 possessed the highest density, which can reach 3.17 ± 0.03 g/cm^3^, while the densities of CMC-2 and CMC-3 were only 3.07 ± 0.03 g/cm^3^ and 2.90 ± 0.01 g/cm^3^, respectively. An evident truth can be found, that is, that the density of the C/C-SiC-(Zr_x_Hf_1−x_)C composites is inversely proportional to that of the porous C/C composites. The porous C/C composite (C/C-1) with higher porosity possessed more space to hold the Si-Zr-Hf melt during the infiltration process, which played an important role in contributing weight and increasing density to CMC-1. In addition, the corresponding porosities of the C/C-SiC-(Zr_x_Hf_1−x_)C composites were measured to be 6.76 ± 0.47%, 3.66 ± 0.28% and 3.82 ± 0.51%, which indicated an excellent densification efficiency. During the RMI process, the infiltration efficiency is particularly related to the capillary force and connectivity of the pores in the porous C/C composites. Acting as the infiltration path of the metal melt, the connected pores absorb the metal melt by the capillary force, which can be calculated by the Washburn equation [38]:(4)$$P_{c}=\frac{4\sigma cos\theta }{d},$$
where $P_{c}$ is the capillary force, σ is the surface tension of melt, *d* is the pore diameter and θ is the contact angle. It is evident that the capillary force is inversely proportional to the pore diameter. Regardless of other factors, the smaller the pore diameter, the larger the capillary force. It can be confirmed that the refinement of the pore structure for the C/C-2 and C/C-3 composites possesses a vital catalytic role for the CMC-2 and CMC-3 composites in elevating the capillary force, and then resulting in the increase in densification.

### 3.3. Ablation Properties of C/C-SiC-(Zr_x_Hf_1−x_)C Composites

Figure 6a shows the ablation properties of the C/C-SiC-(Zr_x_Hf_1−x_)C composites after ablation for 60 s. It can be seen that both of the mass and linear ablation rates of the CMC-1 present minimum values compared with those of CMC-2 and CMC-3. CMC-1 exhibits significant ablation resistance with mass and linear ablation rates of 2.696 mg/s and −0.814 µm/s, respectively. The negative linear ablation rate indicates a slight volumetric expansion of CMC-1 parallel to the flame direction after ablation. Under the same ablation condition, the mass and linear ablation rates of CMC-2 are 4.499 mg/s and 0.310 µm/s, respectively, while those of CMC-3 are 7.071 mg/s and 3.449 µm/s, respectively, which is significantly higher than those of CMC-1. The degradation in ablation resistance may be directly related to the decreased content of antioxidant substances. In addition, it can also prove that the (Zr_x_Hf_1−x_)Si_2_ solid solution possesses outstanding ablation resistance in the present test environment. Figure 6b shows the surface temperature curves of the C/C-SiC-(Zr_x_Hf_1−x_)C composites during the ablation process. The surface temperature range can be divided into three classical stages according to the temperature rising rate. Within the initial stage, the surface temperature can rise to about 1600 °C in a very short time, as all of the sample surfaces withstood severe thermal shock and exhibited excellent thermal shock resistance. As time went on, the surface temperature of the all samples increased gradually in the second stage and then reached a steady state in the third stage. It is worth noting that the surface temperature presents a significant difference in the two following stages. It can be seen from Figure 6b that both of the process temperature and steady-state temperature for different C/C-SiC-(Zr_x_Hf_1−x_)C composites represent the same correspondence. The surface peak temperatures of the C/C-SiC-(Zr_x_Hf_1−x_)C composites were 1799 °C, 1935 °C and 2091 °C. It is well known that a higher temperature can lead to more serious ablation and material consumption, such as oxidation, volatilization or decomposition, which may result in the higher ablation rates of CMC-2 and CMC-3. In addition, the different ceramic content and composition lead to the changes in thermal conductivity and type of chemical reaction, which may also be responsible for the various ablation properties of the C/C-SiC-(Zr_x_Hf_1−x_)C composites.

### 3.4. Ablation Behavior of C/C-SiC-(Zr_x_Hf_1−x_)C Composites

Figure 7 shows the XRD patterns of the C/C-SiC-(Zr_x_Hf_1−x_)C composites after ablation for 60 s. It can be seen that the diffraction peaks of the ZrO_2_ and HfO_2_ phases can be detected from all ablation surfaces, which resulted from the oxidations of the (Zr_x_Hf_1−x_)C or (Zr_x_Hf_1−x_)Si_2_ solid solutions during the ablation process. The SiC phase may originate from its own incomplete oxidation. In addition, the absence of C and SiO_2_ peaks in the XRD patterns can be attributed to the combustion of carbon phase and the amorphous state of the SiO_2_ phase. During the ablation process, the Si-containing glassy phases can effectively spread on the sample surface, as a result of their better liquidity and the erosion of gas flow. The rapid cooling may induce the glassy phases laying on the surface in an amorphous form, which has been confirmed in related studies [6,39].

Figure 8 shows the photographs of the C/C-SiC-(Zr_x_Hf_1−x_)C composites before and after ablation. It can be seen that all of the C/C-SiC-(Zr_x_Hf_1−x_)C composites exhibit good structure integrity, although some exposed carbon phase has been burned out during the ablation process, as shown in Figure 8b,c. Among the CMC composites, CMC-1 exhibits the smoothest ablation surface without visible structural damage. Conversely, CMC-2 and CMC-3 seem to suffer from more serious ablation. A significant number of strip-shaped ablation pits can be observed on the ablation surface, which can provide direct evidence for their higher mass and linear ablation rates. In addition, some white oxides can be observed on the central region of the ablation surface, which plays an important role in protecting the C/C-SiC-(Zr_x_Hf_1−x_)C composites from continuous ablation. Thus, all C/C-SiC-(Zr_x_Hf_1−x_)C composites prepared in this work exhibit an excellent ablation resistance.

Figure 9 shows the cross-section SEM micrographs and EDS results of the C/C-SiC-(Zr_x_Hf_1−x_)C composites in the central region after ablation. It can be seen that the surface layers of the C/C-SiC-(Zr_x_Hf_1−x_)C composites keep good structural stability, and no visible destructive damage can be observed. According to the diffusion depth of oxygen obtained from the scan results, as shown in Figure 9c,f,i, it can be found that the thickness of the oxidation layer for CMC-1 is just 60 μm, which is lower than those of CMC-2 and CMC-3. The high-content ceramics and alloys resulted in the outstanding ablation resistance of the web layers. Based on the EDS analysis (Figure 9j,k) of point 1 and 2 in Figure 9b, the formation of (Zr_x_Hf_1−x_)O_2_ solid solution and the retention of SiC were confirmed. The existence of SiC in the ablation layer may benefit from the preferential oxidation of (Zr_x_Hf_1−x_)C or (Zr_x_Hf_1−x_)Si_2_ phases as well as the running of liquid SiO_2_ during the ablation process. Compelling evidence was provided in Figure 9a according to which (Zr_x_Hf_1−x_)Si_2_ in the ablation layer represented certain chemical stability, and no visible oxidation happened during the ablation process. Thus, it can be inferred that (Zr_x_Hf_1−x_)C possesses much higher oxygen-captured capacity than (Zr_x_Hf_1−x_)Si_2_ and SiC. A preferential oxidation theory of ZrC and HfC had also been proposed in the related studies [40,41]. Relative to CMC-1, the ablation layer of CMC-2 seemed slightly loose, as shown in Figure 9d, in which the integrity of the materials suffered damage to some extent. The continuous SiO_2_ layer, as shown in Figure 9e, can be found on the ablation surface according to the EDS analysis shown in Figure 9l. Moreover, visible cracks can also be observed at the interface between the ablation layer and the non-ablation one, which may result from the volume expansion of (Zr_x_Hf_1−x_)O_2_ during the oxidation process. The delamination failure between the SiO_2_ layer and the underlying (Zr_x_Hf_1−x_)O_2_ particles may be attributed to the thermal mismatch during the cooling process after ablation. As mentioned above, the CMC-3 suffered the highest surface temperature (~2100 °C) during the ablation process. Thus, the CMC-3 contributed to the thickest ablation layer. In particular, a more complex ablation layer was formed, as shown in Figure 9g, in which a significant amount of (Zr_x_Hf_1−x_)O_2_ can be observed clearly, which may indicate the oxidation of (Zr_x_Hf_1−x_)Si_2_. Some visible pores can also be found in the ablation layer, which may result from the rapid escape of gas products. Moreover, a porous oxide structure can be distinguished in the top surface of the ablation layers. Figure 9h shows the enlarged micromorphology of Figure 9g. This region is mainly composed of white (Zr_x_Hf_1−x_)O_2_ and an intermediate mixed phase. The EDS result in Figure 9m shows that the mixed phase is composed of C, O, Si, Zr and Hf elements, which should be a Si-Zr-Hf-O glassy phase formed during the ablation process. In addition, the white (Zr_x_Hf_1−x_)O_2_ appeared under various forms, such as dense-shell, particle-stacked, porous and blocky. These multiform oxides formed in the ablation layer are mainly attributed to the oxidation degree of (Zr_x_Hf_1−x_)C or (Zr_x_Hf_1−x_)Si_2_, as well as the interaction with SiO_2_. Moreover, a dissolving behavior of (Zr_x_Hf_1−x_)O_2_ in glassy phase can also be observed at the adjacent boundary between the (Zr_x_Hf_1−x_)O_2_ and Si-Zr-Hf-O glassy phase. In summary, the ablation surface of CMC-3 was covered by a dense oxide film, which can effectively inhibit the diffusion of oxygen, withstand the scouring of combustion flow and improve the ablation resistance of the C/C-SiC-(Zr_x_Hf_1−x_)C composites.

Figure 10 shows the surface SEM micrographs and EDS results of the C/C-SiC-(Zr_x_Hf_1−x_)C composites in the central region after ablation. It can be seen from Figure 10a that there is no serious ablation damage in the central ablation region of CMC-1. On the ablation surface, part of the exposed (Zr_x_Hf_1−x_)C has been oxidized into a porous form, as shown in Figure 10b, and some cracks between the adjacent oxides have been filled by molten oxides. In addition, a solidified SiO_2_ glassy phase can be observed clearly on the surface of (Zr_x_Hf_1−x_)O_2_ and residual SiC based on the EDS analyses (Figure 10j,k) of specific areas labeled in Figure 10c. The gray-white phase can be identified as (Zr_x_Hf_1−x_)O_2_ covered by the SiO_2_ glassy phase, and the dark-gray phase is SiO_2_-covered SiC, which indicates the formation of molten SiO_2_ on the surface of CMC-1 during the ablation process. The CMC-2 with more severe ablation damage possesses a rougher ablation surface than that of the CMC-1, as shown in Figure 10d, and visible ablation pits resulting from the burn of the carbon phase can be detected. In addition, the residual SiC phase can also be found on the ablation surface. It can be seen from Figure 10e that spherical SiO_2_ is adhered onto the surface of the continuous glassy phase, and defects such as pores and cracks can be observed on it. The spherical SiO_2_ was mainly formed by the vapor deposition of SiO or the high-temperature oxidation of siliceous components. As shown in Figure 10f, the dense-shell (Zr_x_Hf_1−x_)O_2_ and Si-Zr-Hf-O glassy phase occupy most of the surface, and this special oxide film exhibits a dense structural characteristic. The pores in the oxide film act as the main channels for the release of gas products. The dense ablation surface of CMC-3 can be observed, unlike for CMC-1 and CMC-2, as shown in Figure 10g. The oxidation products with a lower melting point melted and spread on the ablation surface under the temperature of almost 2100 °C, and the central ablation region was well protected by the continuous oxide film. As shown in Figure 10h, the (Zr_x_Hf_1−x_)O_2_, SiO_2_ and Si-Zr-Hf-O phases build a diffusion barrier to isolate the oxidation atmosphere. In addition, the Si-Zr-Hf-O glassy phase is usually found between the (Zr_x_Hf_1−x_)O_2_ phase and the SiO_2_ phase, so it is reasonable to believe that the Si-Zr-Hf-O glassy phase was formed as the product of them. In the (Zr_x_Hf_1−x_)O_2_ aggregation area, as shown in Figure 10i, dense-shell (Zr_x_Hf_1−x_)O_2_ and clusters of (Zr_x_Hf_1−x_)O_2_ can be observed. Their formation can be explained by the adsorption theory at liquid interfaces [42,43], which has been reported elsewhere [44]. According to the EDS result in Figure 10l, (Zr_x_Hf_1−x_)O_2_ is surrounded by the dense glassy phase. It can be inferred that a liquid–solid two-phase structure was formed in this region during the ablation process, namely solid (Zr_x_Hf_1−x_)O_2_ and liquid glassy phase. The dual characteristics of oxidation and erosion resistance can ensure the ablation resistance of the C/C-SiC-(Zr_x_Hf_1−x_)C composites effectively.

The ablation of the C/C-SiC-(Zr_x_Hf_1−x_)C composites includes a series of complex physical and chemical processes. Due to their heterogeneous structure, all of the constituent substances such as carbon fiber, PyC matrix, carbides as well as alloys are exposed in the ablation environment. During the ablation process, the exposed carbon phase will be rapidly consumed until the appearance of underlying antioxidant components. Subsequently, visible ablation pits would be left at the corresponding positions. At the same time, (Zr_x_Hf_1−x_)C started to oxide prior to the SiC and (Zr_x_Hf_1−x_)Si_2_ phases, resulting in the formation of porous (Zr_x_Hf_1−x_)O_2_ products. In addition, a small amount of SiO_2_ was formed and melted during the ablation process, and then spread onto the ablation surface or infiltrated into the porous (Zr_x_Hf_1−x_)O_2_, which effectively prevented the further diffusion of oxygen. As ablation proceeds, more and more oxides were formed on the ablation surface, while the reaction between solid (Zr_x_Hf_1−x_)O_2_ products and liquid SiO_2_ as well as the oxidation of (Zr_x_Hf_1−x_)Si_2_ promoted the formation of the Si-Zr-Hf-O phase with a low melting point. The Si-Zr-Hf-O phase can form a continuous oxide film together with SiO_2_, namely the Bi-liquid phase, which possesses an excellent barrier effect. The formation of the Bi-liquid phase can inhibit the rapid volatilization of SiO_2_ at high temperatures, which was also beneficial to ensure the ablation resistance. Moreover, the liquid–solid two-phase structure can play a significant role in retarding the further ablation of the C/C-SiC-(Zr_x_Hf_1−x_)C composites as well as reducing the erosion loss of the liquid glassy phases.

## 4. Conclusions

C/C-SiC-(Zr_x_Hf_1−x_)C composites were prepared by the CVI and RMI methods. The as-prepared CMC composites exhibited a dense structure and were mainly composed of carbon fiber, carbon matrix, SiC ceramic, (Zr_x_Hf_1−x_)C and (Zr_x_Hf_1−x_)Si_2_ solid solutions. The refinement of the connected pores plays a vital catalytic role in elevating the capillary force and promoting the conversion from metal to carbide. The C/C-SiC-(Zr_x_Hf_1−x_)C composites exhibited excellent ablation properties under plasma flame. After ablation for 60 s, the minimum mass and linear ablation rates were only 2.696 mg/s and −0.814 µm/s, respectively. The structural difference is the key factor influencing the ablation properties and behavior of the C/C-SiC-(Zr_x_Hf_1−x_)C composites. During the ablation process, a Bi-liquid phase consisting of SiO_2_ and Si-Zr-Hf-O phases covers the ablation surface, which can act as an oxygen diffusion barrier to retard the underlying components from further ablation. Meanwhile, the formation of a liquid–solid two-phase structure was beneficial to entrust the excellent ablation properties of the C/C-SiC-(Zr_x_Hf_1−x_)C composites.

## Figures and Tables

**Figure 1 materials-16-02120-f001:**
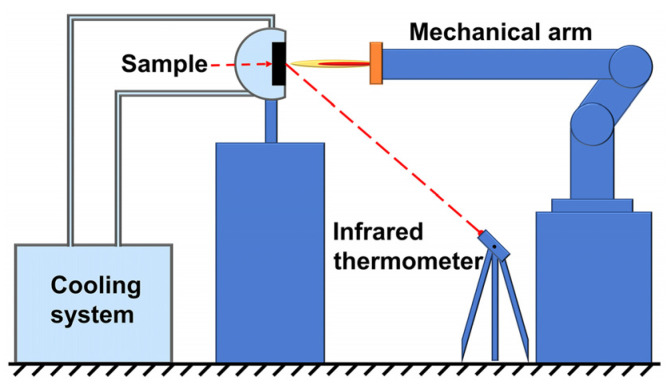
Diagram of air–plasma ablation device.

**Figure 2 materials-16-02120-f002:**
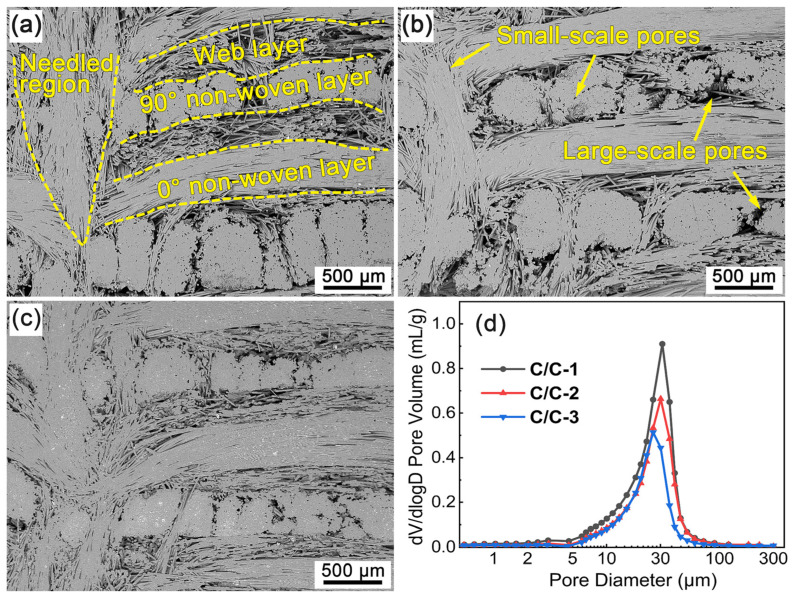
Cross-section SEM micrographs and pore size distributions of the porous C/C composites: (**a**) C/C-1; (**b**) C/C-2; (**c**) C/C-3; (**d**) Pore size distribution.

**Figure 3 materials-16-02120-f003:**
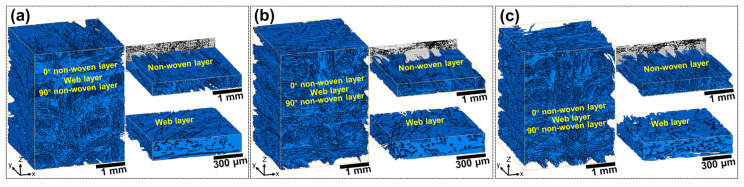
3D reconstruction graphs of connected pores of the porous C/C composites with various densities: (**a**) C/C-1; (**b**) C/C-2; (**c**) C/C-3.

**Figure 4 materials-16-02120-f004:**
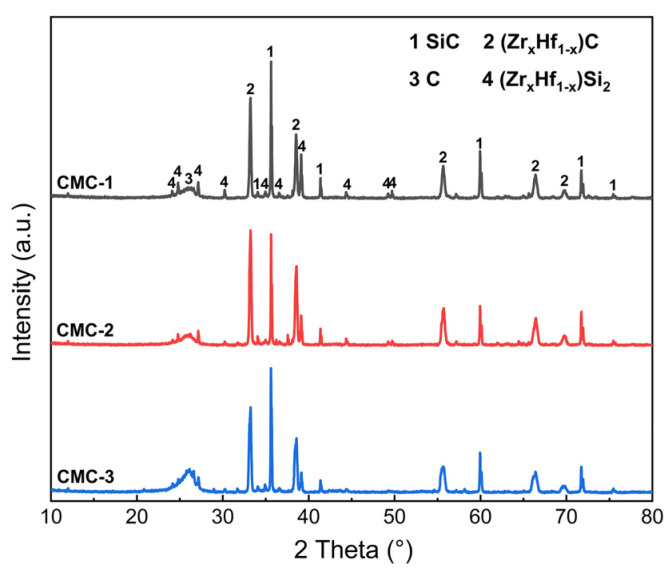
XRD patterns of the CMC composites prepared by RMI method.

**Figure 5 materials-16-02120-f005:**
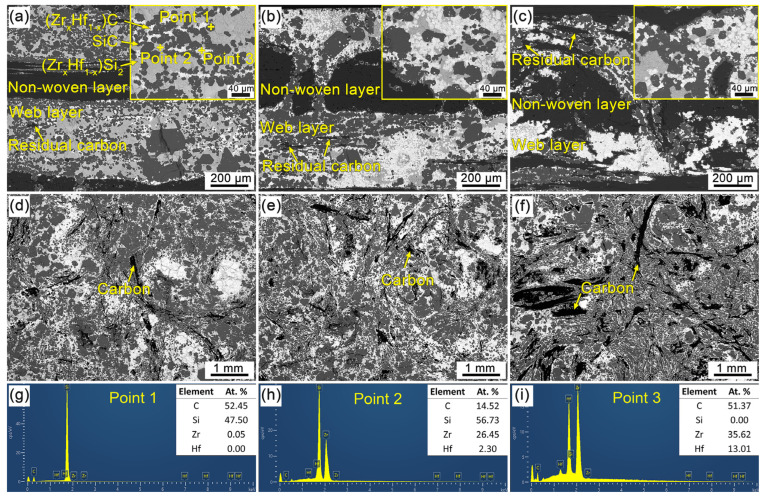
SEM micrographs and EDS results of the C/C-SiC-(Zr_x_Hf_1−x_)C composites: (**a**,**d**) CMC-1; (**b**,**e**) CMC-2; (**c**,**f**) CMC-3; (**g**–**i**) EDS results.

**Figure 6 materials-16-02120-f006:**
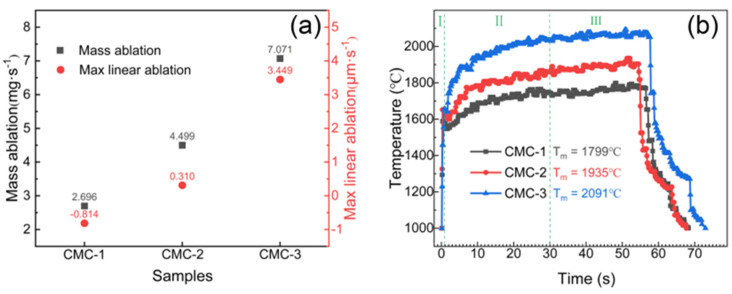
Ablation rates and surface temperature curves of C/C-SiC-(Zr_x_Hf_1−x_)C composites: (**a**) Ablation properties; (**b**) Surface temperature curves.

**Figure 7 materials-16-02120-f007:**
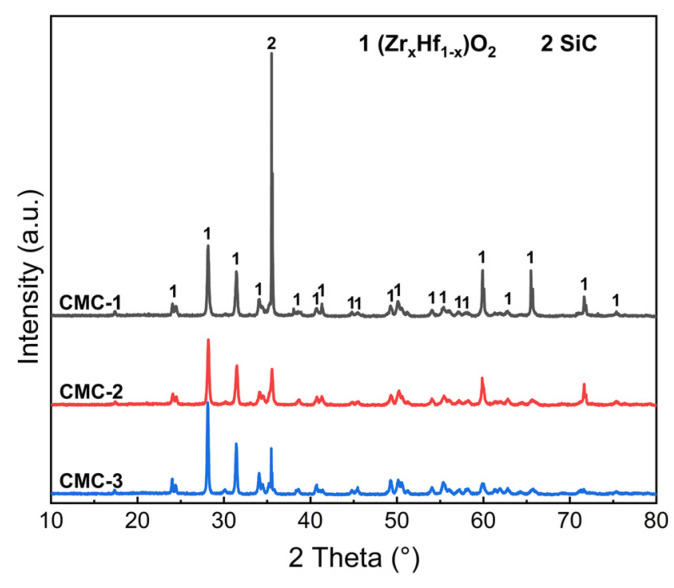
XRD patterns of the C/C-SiC-(Zr_x_Hf_1−x_)C composites after ablation.

**Figure 8 materials-16-02120-f008:**
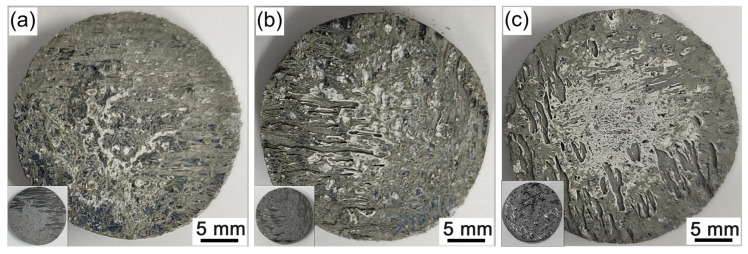
Photographs of the C/C-SiC-(Zr_x_Hf_1−x_)C composites before and after ablation: (**a**) CMC-1; (**b**) CMC-2; (**c**) CMC-3.

**Figure 9 materials-16-02120-f009:**
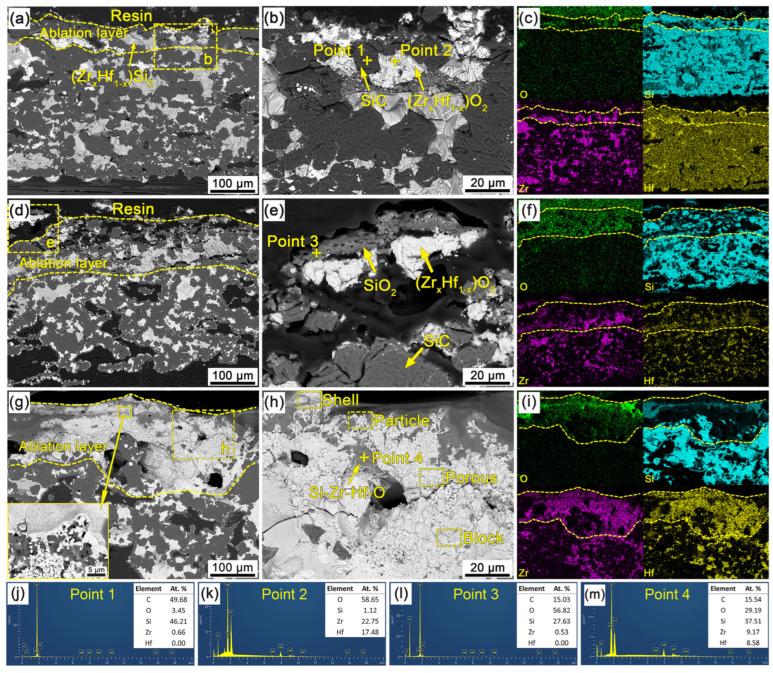
Cross-section SEM micrographs and EDS results of the C/C-SiC-(Zr_x_Hf_1−x_)C composites in the central region after ablation: (**a**–**c**) CMC-1; (**d**–**f**) CMC-2; (**g**–**i**) CMC-3; (**j**–**m**) EDS results.

**Figure 10 materials-16-02120-f010:**
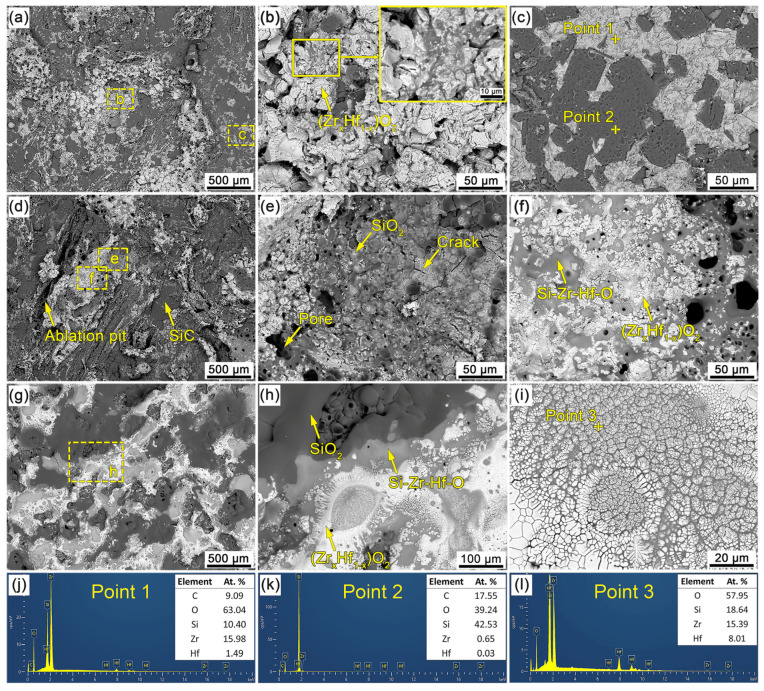
Surface SEM micrographs and EDS results of the C/C-SiC-(Zr_x_Hf_1−x_)C composites in the central region after ablation: (**a**–**c**) CMC-1; (**d**–**f**) CMC-2; (**g**–**i**) CMC-3; (**j**–**l**) EDS results.

**Table 1 materials-16-02120-t001:** The structural parameters of the connected pores in the porous C/C composites.

Sample	Fractal Dimension	Z-Direction Tortuosity
C/C-1	2.68	1.73
C/C-2	2.62	1.85
C/C-3	2.62	1.88

## Data Availability

Data available upon request from the authors.

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
