# Peer review of "Microstructure and Ablation Behavior of C/C-SiC-(ZrxHf1−x)C Composites Prepared by Reactive Melt Infiltration Method"

_materials, 2023, doi:10.3390/ma16052120_

Round 1
Reviewer 1 Report
The presentation and overall writing of the manuscript were found very good. The topic was very interesting and timely, since currently an emphasize is given on developing space materials. The following comments can be addressed in the updated manuscript to improve the quality of the manuscript further-
1. Line 33, please specify the reference and information one by one , not just put all of the references together, so that it can be understood which information was collected from which reference sources.
2. Line 42, please expand the mechanism of this sentence, not very clear.
3. Line 45, glassy should start with capital G.
4. Please provide details of the findings of reference 18 in line 56-60, so that the novelty aspect of this work can be understood and expressed better.
5. Materials and methods- author can prefer to use sub-section/sub-headings to guide readers about materials/manufacturing/different characterisation processes.
6. Materials and methods- line 83-92- many characterisation processes were mentioned very briefly. A good details ( and one by one) should be given about the characterisation processes so that any future investigation by other researchers they can be repeated.
7. For the ablation testing process- how the parameters were selected? is there any test standard available for this testing? or any previous references was followed?
8. How the pore size, volume and interconnectivity were measured in the X-ray u-CT test.?
9. For frig. 4 X-rd data, references should be cited when different peaks were explained.
10. In Figure 5i, why CMC 3 showed no Si in the EDS analysis, please explain.
11. Line no. 311- Fig 8 didn't provide any CMC images before ablation testing.
Reviewer 2 Report
The article clearly laid out with all essentials listings viz. abstract, introduction, methodology, results, conclusions. The title of the article is well suited for the presented study. The introduction summarizes relevant research and besides clearly describes the hypotheses and experimental methods used in. The authors have used a well-suited methodology for collecting data for their intended studies. The findings from various studies are very well organized and presented in the results part. Discussions are made in the light of the obtained results. Finally, the authors have successfully made reasonable conclusions from their studies. Throughout the article, the authors have maintained a reasonable consistency in presenting the text, figures, tables etc.
Finally, I recommend this article for possible publication in your esteemed journal after the following minor correction:
Check the XRD patterns Fig.4 and Fig.7
Reviewer 3 Report
Comment
In this paper the author claimed to focus on the Microstructure and ablation behavior of C/C-SiC-(ZrxHf1-x)C 2 composites prepared by the reactive melt infiltration method. The project target C/C-SiC-(ZrxHf1-x)C composites which were prepared by chemical vapor infiltration and reactive melt infiltration methods. The microstructure and ablation behavior of the C/C-SiC-(ZrxHf1-x)C composites were systematically investigated. The results show that the characteristics of connected pores possess significant effects on the microstructure and ablation resistance of the C/C-SiC- (ZrxHf1-x)C composites. After ablation for 60 s, the minimum mass and linear ablation rates are only 2.696 mg/s and -0.814 µm/s, respectively. During the ablation process, a Bi-liquid phase and a liquid-solid two-phase structure were formed, which are responsible for the excellent ablation resistance of the C/C-SiC-(ZrxHf1-x)C composites. The paper stands a chance of adding to existing knowledge provided the authors can provide answers to the following comments.
1. The author should discuss the reason for the sharp peak increase in Silicon, SiC observed in the sample designated as CMS-1 in Figure 7 of XRD patterns of the C/C-SiC-(ZrxHf1-x)C composites after ablation.
2. The author should reacquire the microstructure in figure 10 to ensure the same uniformity in the scale bar.
3. The statement in lines 409 -410 should be revised by the authors.
4. The milling balls used should be clearly stated with reasons. How did the authors ensure the prevention of cold welding because of the prolonged milling time?
5. The authors should indicate the parameters used during the infiltration process with references to literature in order to clearly justify the parameters mentioned.
6. The whole abstract section needs to be rewritten to actively convey the purpose of the manuscript.
7. The discussion made in the manuscript needs to be visited with additional technical points which is detailed enough to fully justify the experimental results in the manuscript which is capable of making the idea publishable.
8. The concluded remark is fully justified in the main work.
